# Use of *Lachancea thermotolerans* for Biological vs. Chemical Acidification at Pilot-Scale in White Wines from Warm Areas

Cristian Vaquero [1], Pedro Miguel Izquierdo-Cañas [2,3], Adela Mena-Morales [2], L. Marchante-Cuevas [2], José María Heras [4] and Antonio Morata [1,*]

1   enotecUPM, Chemistry and Food Technology Department, ETSIAAB, Universidad Politécnica de Madrid, Avenida Puerta de Hierro 2, 28040 Madrid, Spain; c.vaquero@upm.es
2   Instituto Regional de Investigación y Desarrollo Agroalimentario y Forestal de Castilla-La Mancha (IRIAF), IVICAM, Ctra. Albacete s/n, 13700 Tomelloso, Spain; pmizquierdo@jccm.es (P.M.I.-C.); amenam@jccm.es (A.M.-M.); lmarchantec@jccm.es (L.M.-C.)
3   Parque Científico y Tecnológico de Castilla-La Mancha, Paseo de la Innovación 1, 02006 Albacete, Spain
4   Lallemand Ibérica, 28521 Madrid, Spain; jmheras@lallemand.com
*   Correspondence: antonio.morata@upm.es

**Abstract:** Climate change is affecting vineyards, resulting in grapes with a low acidity a high pH and sugar at harvest time. The most common procedure so far to improve the acidity and reduce the final pH of wines is to use tartaric acid, but wine can also be acidified microbiologically using *Lachancea thermotolerans* yeasts, a natural bio-tool that acidifies gradually during the first stage/days of fermentation. Two strains of *L. thermotolerans* were compared with one *Saccharomyces cerevisiae* at a pilot-scale under similar fermentation conditions and in duplicate. A sequential inoculation was performed on the third day for the non-*Saccharomyces*, producing only about 1 g/L of lactic acid, which was suitable for comparison with the *Saccharomyces*, to which 1.5 g/L of tartaric acid had been added to lower the final pH. The three fermentations ended with a total acidity without significant differences. A significant and normal feature of the *L. thermotolerans* yeasts is their higher propane-1,2,3-triol production, which was observed in the Laktia yeast, and the acetic acid was <0.3 g/L. The amount of volatile metabolites was generally higher for non-*Saccharomyces* and the increase was seen in carbonyl compounds, organic acids, lactones, fumaric compounds, and phenols. Finally, the sensory analysis showed that there were hardly any significant differences, even though the non-*Saccharomyces* had a higher quantity of volatile metabolites, which could lead to a good acceptance of the product, since biological acidification was used, generating a more natural product.

**Keywords:** acidity; natural bio-tool; *Lachancea thermotolerans*; *Saccharomyces cerevisiae*; tartaric acid; volatile profile; pilot-scale

## 1. Introduction

Winemakers are increasingly realising the importance of using more than one type of yeast for their fermentations and are including non-*Saccharomyces* yeasts to generate a higher quantity and a greater diversity of metabolites in their wines [1–3]. Among the most studied are *Hanseniaspora* spp. [4–6], *Metschnikowia pulcherrima* [7,8], *Torulaspora delbrueckii* [9,10], *Lachancea thermotolerans* [11,12], *Schizosaccharomyces pombe* [13,14], and *Candida stellata* [15] not only in wines but also in beers [16,17]. One of the non-*Saccharomyces* yeasts that increases both the acidity of wines and the amount of volatile metabolites is *L. thermotolerans* [11,18]. This ubiquitous yeast [19] is increasingly being used because global warming causes grapes with high sugar levels, a very low acidity, and a high pH, resulting in fermentations with a very high alcohol content, leading to a fermentation process that is not completely finished and, in general, leads to a sensory imbalance [20]. The most commonly used solutions involve treating the wines with tartaric acid or using cation exchangers [21,22]. This yeast, which cannot finish fermentation on its own and

must be used in co-inoculation or, better still, in sequence with *S. cerevisiae* [1,23,24] can produce, depending on the strains, up to 16.6 g/L of lactic acid and can reach 2–3 g/L total acidity and a pH reduction of 0.3 when the co-inoculation ratio is 7-log/3-log CFU/mL of *L. thermotolerans*/*S. cerevisiae* [25,26]. This is the case even on an industrial scale, where the differences between the populations of the two yeasts are less than one log unit and can reach more than 2 g/L of lactic acidification [23]. This acidity production is due to the expression of lactate dehydrogenase (LDH) genes from pyruvate derived from glycolysis [27].

Among other characteristics, *L. thermotolerans* produces its highest amount of lactic acid between day 4 and 6 of fermentation, depending on how competitive it is with other yeasts [11,12]. This lactic acid production can come from berry sugars, which leads to a decrease of 0.3–0.7% vol of the final ethanol in the wine [28]. There is an increasing demand for these low alcohol wines in the market [29]. *L. thermotolerans* is a yeast that rarely generates or tolerates more than 9–10% vol of ethanol and also rarely produces more than 0.3–0.5 g/L of volatile acidity [18,30]. It can also consume acetic acid through its respiration metabolism [31] and can generate more propane-1,2,3-triol [32]. On the other hand, the importance of this yeast in the production of aromatic and fruity esters with citrus and non-dairy flavours should be highlighted [12,23,33], and these values can be further enhanced if the must is oxygenated daily at the beginning of fermentation, as it improves the growth of *L. thermotolerans* [34]. All these results lead to wines with better aromas, flavours, and mouthfeel, as well as a greater freshness [35,36].

In the present study, the aim was to observe the oenological potential at a pilot-scale of the *Lachancea thermotolerans* (Lt) strain L31 (enotecUPM, ETSIAAB, UPM, Madrid, Spain) and LEVEL2 LAKTIA™ (Lallemand, Montreal, QC, Canada), with a *Saccharomyces cerevisiae* control strain Lalvin QA23™ which was selected in the Vinhos Verdes region of Portugal by Lallemand.

## 2. Materials and Methods

### 2.1. Yeast Used

The following active dry non-*Saccharomyces* and *Saccharomyces* cerevisiae yeast strains were used in the fermentations: *Lachancea thermotolerans* (Lt) strain L31 (Bodegas Comenge and enotecUPM. Valladolid, Spain), strain Laktia (Lallemand. LAKTIA™) (LKT), and *Saccharomyces cerevisiae* strain Lalvin QA23 (Lallemand. LALVIN QA23™) (QA23), which was used as the control.

### 2.2. Fermentation Trials

Sequential fermentation was carried out on a pilot-scale in 30 L stainless steel tanks to test the performance of the different strains of *Lachancea thermotolerans* by determining the fermentation kinetics and the fermentative power.

A manual harvest was carried out (Supplementary Table S1) to obtain a white must (*Vitis vinifera* L. cv. Airén) with 20.95 °Brix, a pH of 4.05, sugars at 197 g/L, a yeast assimilated nitrogen (YAN) of 179 mg/L, and a total acidity of 2.67 g/L. The must was destemmed, and 1 g of $SO_2$ + 1.5 g of ascorbic acid/100 kg grapes was added. After pneumatic pressing, 1 g/hL of $SO_2$ + 2 g/hL of ascorbic acid + 30 g of *Glutastar*™/100 kg grapes was added to the musts for further protection against oxidation. It was then left to settle for 24 h with the addition of 1 g/hL of *Lallzyme HC*™ for static settling and with a turbidity after decanting of 25.3 NTU. The musts were inoculated with 25 g/hL each of *L. thermotolerans* and Q23 at 18 °C in duplicate, with volumes of 24 L per tank in a total of six tanks. Sequential inoculation (Laktia→QA23, L31→QA23) was performed at 72 h with 25 g/hL of Q23 yeast. At the beginning of fermentation, 1.5 g/L of tartaric acid was added to the musts with the Q23 yeast, and 30 g/hL of *Go-Ferm*® *Protect* was added to all fermentations when necessary for yeast microprotection and 20 g/hL of *Nutrient Vit Blanc*™ as a nutrient deficiency corrector for the must. Fermentation lasted for 14 days.

*2.3. Oenological Parameters of Wines*

The principal oenological parameters were analysed according to the official analytical methods [37]. These included alcoholic content, total acidity, pH, total and free SO$_2$, glucose and fructose, propane-1,2,3-triol, different acid concentrations, and the Folin–Ciocalteu method, and the chromatic characteristics were analysed at pH 3.6 using the CIELAB colour space [38].

*2.4. Analysis of Fermentative Volatile Compounds Using Gas Chromatography-Mass Spectrometry (GC-MS)*

Volatile compounds were obtained by GC-MS on a FocusGC system coupled to a model ISQ mass spectrometer (electron impact ionisation source and quadrupole analyser) equipped with a TriPlus autosampler (ThermoQuest, Waltham, MA, USA), with a BP21 column (50 m × 0.32 mm × 0.25 μm of the free fatty acid phase) (SGE, Ringwood, Australia). The detector had the following conditions: impact energy, 70 eV; electron multiplier voltage, 1250 V; ion source temperature, 250 °C; and mass scan range, 40–250 amu. The chromatographic conditions were: carrier helium gas (1.4 mL/min, 1/57 split); injector temperature, 220 °C; and oven temperature, 40 °C for 15 min, 2 °C min$^{-1}$ at 100 °C, 1 °C/min at 150 °C, 4 °C/min at 210 °C, and 55 min at 210 °C. 4-nonanol (0.1 g/L) and an SPE cartridge (LiChrolut EN, Merck, 0.2 g phase, Darmstadt, Germany) were used as internal standards following the methodology developed by the authors of [39]. Volatile compounds were quantified by *m/z* fragments that were characteristic for each compound using the internal standard method. The concentration of the unavailable compounds was expressed in μg/L or mg/L as 4-nonanol equivalents.

*2.5. Sensory Analysis*

A panel of 18 experienced tasters (aged between 30 and 65 years) evaluated the wines that had been bottled and kept under refrigeration for three months. The blind tasting took place in the tasting rooms of the Instituto Regional de Investigación y Desarrollo Agroalimentario y Forestal de Castilla-La Mancha (IRIAF), and in the Department of Chemistry and Food Technology of the Polytechnic University of Madrid, which were equipped with fluorescent lighting, and the samples were presented in random order. The wines (30 mL/tasting glass) were served at 12 ± 2 °C in standard, odourless tasting glasses. A glass of water was also provided to the panellists to clean their palates between samples. Attributes were chosen for sensory analysis, with three visual, six for aroma and nine for taste. Panellists used a scale of 1 to 5 to rate the intensity of each attribute. On the scale, 0 represented "non-perceptible attribute" and 5 represented "strongly perceptible attribute". Each panellist also evaluated the overall impression, taking into account olfactory and gustatory aspects, as well as the lack of defects. The tasting sheets also had a final blank space for any additional comments or observations on the sensory notes or nuances not previously included as attributes.

*2.6. Statistical Analysis*

SPSS version 23.0 software (SPSS Inc., Chicago, IL, USA) was used to perform a one-way analysis of variance (ANOVA) (Student–Newman–Keuls test, α = 0.05) to compare the wine data for each of the duplicates considered. A one-way ANOVA and a Tukey's test (α = 0.05) were performed to highlight the differences between the treatments for sipping. A principal component analysis (PCA) was performed using Addinsoft (2021), XLSTAT statistical and data analysis solution, New York, USA.

**3. Results**

*3.1. Oenological and Fermentation Parameters*

Fermentations started 48 h after inoculation (Figure 1). At 72 h, *Saccharomyces* (QA23) was inoculated, with a rapid drop in density occurring almost equally/uniformly in the fermentations. The Laktia→QA23 yeast had a slightly faster fermentation until day 5,

where the tumultuous fermentation decreased with all yeasts to 1010.5 ± 1.6. Both the QA23 and L31→QA23 yeast finished their fermentation on day 9 (991 ± 0.0) and the Laktia→QA23 yeast was left until day 12. No variations were seen in their analytical results (993 ± 2.1).

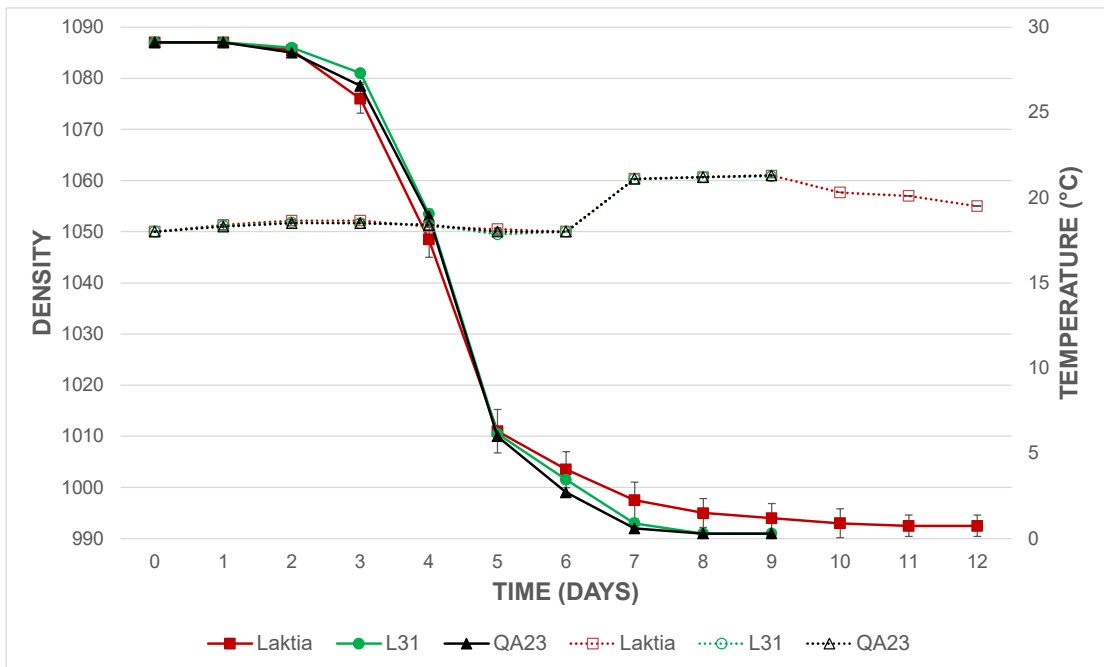

**Figure 1.** Comparative evolution of density and temperature during must fermentation.

On the other hand, as the tanks were not thermally controlled but were placed in a temperature-controlled pool, it was possible to record the fermentation temperatures, which started at 18 °C and, until day 6, were around 18.3 ± 0.1 °C. From that day on, there was an exponential rise to 21.2 ± 0.0 °C, and in the case of Laktia→QA23, from day 9 onwards, this temperature began to gradually decrease.

Regarding the oenological parameters (Table 1), there were significant differences in ethanol and not in the residual sugars in Laktia→QA23. The pH was slightly higher in L31→QA23. The acetic acidity was always lower than 0.4 g/L. Laktia→QA23 yeast produced the most propane-1,2,3-triol. There were significant differences in the two *L. thermotolerans* yeasts compared to *S. cerevisea* yeast in lactic acid production.

**Table 1.** Oenological parameters of the different yeasts on the last day of fermentation. Different letters in the same row indicate statistically significant differences according to the Student–Newman–Keuls (S–N–K) test. α = 0.05.

|  | **Laktia→QA23** | **L31→QA23** | **QA23** |
|---|---|---|---|
| Ethanol (% *v/v*) | 12.29 ± 0.08 a | 12.75 ± 0.07 b | 12.81 ± 0.12 b |
| Total acidity (g/L) | 4.31 ± 0.51 a | 4.18 ± 0.17 a | 3.95 ± 0.10 a |
| pH | 3.74 ± 0.07 a | 3.81 ± 0.03 b | 3.72 ± 0.01 a |
| Total SO$_2$ (mg/L) | 89 ± 10 a | 110 ± 35 a | 100 ± 20 a |
| Free SO$_2$ (mg/L) | 10 ± 4 a | 21 ± 13 a | 20 ± 9 a |
| Glucose/Fructose (g/L) | 1.65 ± 1.39 a | 0.87 ± 0.74 a | 0.31 ± 0.04 a |
| Acetic acid (g/L) | 0.18 ± 0.03 a | 0.17 ± 0.00 a | 0.25 ± 0.00 b |
| Malic acid (g/L) | 1.51 ± 0.11 a | 1.56 ± 0.04 a | 1.62 ± 0.02 a |
| Lactic acid (g/L) | 1.02 ± 0.22 b | 0.90 ± 0.11 b | 0.07 ± 0.10 a |
| Citric acid (g/L) | 0.07 ± 0.00 a | 0.06 ± 0.00 a | 0.06 ± 0.00 a |
| Succinic acid (g/L) | 0.81 ± 0.02 b | 0.73 ± 0.0 a | 0.82 ± 0.01 b |
| Tartaric acid (g/L) | 1.10 ± 0.00 a | 1.14 ± 0.07 a | 1.15 ± 0.04 a |
| Propane-1,2,3-triol (g/L) | 6.81 ± 0.11 c | 5.69 ± 0.23 a | 6.11 ± 0.00 b |

### 3.2. Flavonoid Compounds and Colour

Catechins were also analysed (Table 2) at the end of fermentation and a slightly lower amount was observed in the Laktia→QA23 yeast. As for the quantification of total polyphenols in the wine, the Folin–Ciocalteu method was used, showing that the Laktia→QA23 yeast also had lower amount of polyphenols.

**Table 2.** Flavonoid compounds and colour (CIE 1976 L*a*b*) of the different yeasts on the last day of fermentation. Different letters in the same row indicate statistically significant differences according to the Student–Newman–Keuls (S–N–K) test. $\alpha = 0.05$.

|  | **Laktia→QA23** | **L31→QA23** | **QA23** |
|---|---|---|---|
| Catechins (mg/L) | 49.65 ± 5.44 a | 55.35 ± 0.49 b | 54.40 ± 0.14 b |
| Folin–Ciocalteu method (mg GAE/L) | 6.08 ± 0.35 a | 6.62 ± 0.25 b | 6.71 ± 0.05 b |
| L* | 99.229 ± 0.092 a | 99.578 ± 0.196 b | 99.512 ± 0.169 b |
| a* | −1.251 ± 0.037 a | −1.262 ± 0.064 a | −1.166 ± 0.025 b |
| b* | 6.854 ± 0.832 a | 6.639 ± 0.380 a | 6.233 ± 0.271 a |
| Absorbance 420 nm | 0.091 ± 0.010 a | 0.085 ± 0.005 a | 0.080 ± 0.004 a |

On the other hand, the colour of the wine was obtained through CIElab coordinates (CIE 1976 L*a*b*), which is the chromatic model normally used to describe all the colours that can be perceived by the human eye. The L*, which indicates the brightness, was the lowest in Laktia→QA23. The a*, which is negative, tends toward greenish tones, of which QA23 had the lowest value. The b*, which is positive, tends toward yellowish tones, with no significant differences between the yeasts. Finally, the absorbance was at 420 nm, indicating the range of yellow colour and oxidation in the wine, for which no significant differences were noted.

### 3.3. Fermentative Volatiles

The most representative volatiles from the different fermentations (varietal and yeast-derived) were evaluated at the end of fermentation and were organised into esters, alcohols, carbonyl compounds, organic acids, norisoprenoids, terpenes, lactones, fumaric compounds, and volatile phenols (Table 3). Both strains of *L. thermotolerans* strains had a similar trend, although in different proportions.

Esters are compounds responsible for providing fruity aromas in wines. A decrease was observed in almost all esters except for ethyl butyrate and ethyl acetate, which remained the same, and an increase in ethyl lactate by four to six times was observed in *L. thermotolerans*. As for alcohols, in both the *L. thermotolerans* strains, 1-propanol, 3-ethoxy-1-propanol, and trans-3-hexenol decreased, while isobutanol, 1-hexanol, and 3-ethyl-thio-propanol increased, and the rest remained the same. The carbonyl compounds in non-*Saccharomyces* yeasts, acetaldehyde, and acetoin increased. Organic acids decreased in both strains, Laktia and L31, compared to the control QA23, except for isobutyric acid, which increased. As for norisoprenoids and terpenes, there were no major variations except for geraniol, which doubled in quantity in *L. thermotolerans*. In lactones, there was a significant increase in γ-butyrolactone and a slight increase in γ-decalactone in the *L. thermotolerans* strains. In the fumaric compounds, furaneol and hydroxymethylfurfural increased in both the non-*Saccharomyces* strains. Finally, volatile phenols increased 4-vinylphenol and 4-vinylguaiacol and decreased 4-ethylguaiacol in both strains compared to S. cerevisiae. The principal component analysis (PCA) (Figure 2) shows the significant difference between the yeasts used, which can be separated into two groups, discriminated by the ordinate axis, one in the positive part which is the Q23 yeast, and the other which is the Laktia and L31 which is in the negative part of the F2 (green ellipse). Wines fermented with Q23 are characterised by a high amount of esters and alcohols, while Laktia and L31 are characterised by more terpenes, norisoprenoids, and carbonyl compounds.

**Table 3.** Concentration (mg/L and µg/L) of the different volatile compounds analysed at the end of fermentation in the different inoculations with their respective odour thresholds. Different letters in the same row indicate statistically significant differences according to the Student–Newman–Keuls (S–N–K) test. α = 0.05.

| | Threshold Odour | QA23 | Laktia→QA23 | L31→QA23 |
|---|---|---|---|---|
| **ESTERS** | | | | |
| Ethyl acetate (mg/L) | 7.5–12 [40] | 64.62 ± 1.71 a | 66.96 ± 13.95 a | 57.21 ± 2.59 a |
| **Isoamyl acetate (mg/L)** | 0.03 [41] | **2.26 ± 0.16 c** | **0.98 ± 0.21 a↓** | **1.57 ± 0.03 b↓** |
| **Hexyl acetate (µg/L)** | 1500 [41] | **171.60 ± 42.07 b** | **8.20 ± 4.72 a↓** | **14.98 ± 9.37 a↓** |
| **Ethyl lactate (mg/L)** | 150 [42] | **1.79 ± 0.16 a** | **11.76 ± 8.85 b↑** | **7.94 ± 3.96 ab↑** |
| Ethyl butyrate (µg/L) | 20 [41] | 133.50 ± 44.55 a | 93.25 ± 53.39 a | 77.00 ± 53.74 a |
| **Ethyl hexanoate (mg/L)** | 0.005–0.014 [41,43] | **1.31 ± 0.02 c** | **0.60 ± 0.06 a↓** | **0.92 ± 0.11 b↓** |
| **Ethyl octanoate (mg/L)** | 0.6 [41] | **1.31 ± 0.04 c** | **0.57 ± 0.08 a↓** | **1.06 ± 0.07 b↓** |
| **Ethyl decanoate (µg/L)** | 200 [43] | **265.12 ± 28.74 c** | **157.06 ± 0.03 a↓** | **214.98 ± 2.51 b↓** |
| **Diethyl succinate (µg/L)** | 200–500 [43,44] | **229.32 ± 11.32 b** | **191.82 ± 1.83 a↓** | **256.78 ± 33.48 b↓** |
| **Isoamyl lactate (µg/L)** | 200 [45] | **74.38 ± 7.79 c** | **44.45 ± 0.92 a↓** | **60.84 ± 0.58 b↓** |
| **ALCOHOLS** | | | | |
| Isoamyl alcohols (mg/L) | 30–60 [42,43] | 217.10 ± 8.71 a | 252.83 ± 38.29 a | 226.47 ± 2.08 a |
| Methanol (mg/L) | - | 28.03 ± 0.66 b | 28.54 ± 1.45 b | 26.42 ± 0.09 a |
| **1-propanol (mg/L)** | 90–300 [41,42] | **38.16 ± 2.62 c** | **27.27 ± 1.45 b↓** | **14.49 ± 0.22 a↓** |
| **1-butanol (µg/L)** | 150 [42] | **166.00 ± 50.91 a** | **174.50 ± 61.52 a** | **178.00 ± 9.90 a** |
| **Isobutanol (mg/L)** | 40 [46] | **13.22 ± 0.39 a** | **40.50 ± 1.29 c↑** | **25.54 ± 4.01 b↑** |
| **1-hexanol (mg/L)** | 8 [41] | **1.73 ± 0.03 a** | **2.13 ± 0.05 c↑** | **1.98 ± 0.00 b↑** |
| **trans-3-hexenol (µg/L)** | 400–1000 [41,44] | **319.19 ± 5.73 b** | **289.91 ± 13.99 a↓** | **310.16 ± 20.46 ab↓** |
| **3-ethoxy-1-propanol (µg/L)** | 100 [42] | **441.45 ± 11.71 c** | **307.12 ± 39.26 b↓** | **140.54 ± 37.47 a↓** |
| **3-Ethyl-thio-propanol (µg/L)** | - | **4.44 ± 0.06 a** | **10.96 ± 2.82 b↑** | **14.67 ± 1.67 c↑** |
| Benzyl alcohol (µg/L) | 200 [40] | 43.29 ± 1.44 a | 43.19 ± 3.37 a | 47.24 ± 1.95 a |
| 2-phenylethanol (mg/L) | 14 [40] | 34.56 ± 2.18 a | 32.91 ± 0.10 a | 32.71 ± 2.63 a |
| **CARBONYLIC COMPOUNDS** | | | | |
| **Acetaldehyde (mg/L)** | 110 [42] | **28.58 ± 1.57 a** | **48.99 ± 8.49 c↑** | **38.12 ± 2.53 b↑** |
| **3-OH-2-butanone (acetoin) (mg/L)** | 150 [42] | **25.00 ± 12.73 a** | **47.25 ± 2.47 a** | **76.00 ± 25.46 b↑** |
| 2,3-butanedione (diacetyl) (µg/L) | 200–2800 [47] | 168.50 ± 38.89 a | 177.75 ± 25.81 a | 154.00 ± 2.83 a |
| **ORGANIC ACIDS** | | | | |
| **Butyric acid (mg/L)** | 0.17 [41] | **1.66 ± 0.14 c** | **1.14 ± 0.03 a↓** | **1.47 ± 0.07 b↓** |
| **Isobutyric acid (µg/L)** | 2300 [43] | **492.54 ± 2.42 a** | **927.65 ± 131.24 c↑** | **699.46 ± 8.81 b↑** |
| **Isovaleric acid (mg/L)** | 0.03 [41] | **1.74 ± 0.13 b** | **1.34 ± 0.02 a ↓** | **1.29 ± 0.01 a↓** |
| **Hexanoic acid (mg/L)** | 0.42 [41] | **7.63 ± 0.35 c** | **3.50 ± 0.64 a↓** | **5.70 ± 0.25 b↓** |
| **Octanoic acid (mg/L)** | 0.5 [43] | **15.35 ± 0.00 c** | **8.40 ± 1.67 a↓** | **13.03 ± 0.76 b↓** |
| **Decanoic acid (mg/L)** | 1 [43] | **4.26 ± 0.11 c** | **2.77 ± 0.21 a↓** | **3.67 ± 0.05 b↓** |
| **NORISOPRENOIDS** | | | | |
| β-damascenone (µg/L) | 0.05 [40] | 1.22 ± 0.15 a | 1.99 ± 0.79 a | 1.50 ± 0.36 a |
| 3-oxo-α-ionol (µg/L) | - | 17.05 ± 1.84 a | 16.53 ± 2.10 a | 19.63 ± 1.88 a |
| **TERPENES** | | | | |
| Linalool (µg/L) | 25 [40] | 2.66 ± 0.67 a | 3.31 ± 0.28 a | 3.97 ± 2.28 a |
| Citronellol (µg/L) | 100 [41] | 3.05 ± 0.23 a | 3.39 ± 0.26 a | 2.65 ± 0.89 a |
| **Geraniol (µg/L)** | 20–30 [40,41] | **2.99 ± 0.38 a** | **4.47 ± 0.88 b↑** | **4.98 ± 1.51 b↑** |
| **LACTONES** | | | | |
| **γ-butyrolactone (µg/L)** | 35,000 [44] | **91.41 ± 6.85 a** | **225.65 ± 18.27 c↑** | **175.00 ± 6.39 b↑** |
| γ-octalactone (µg/L) | 400 [44] | 38.30 ± 1.67 a | 34.99 ± 0.28 a | 43.00 ± 8.41 a |
| γ-nonalactone (µg/L) | 30 [40] | 4.50 ± 0.46 a | 3,93 ± 1.16 a | 4,22 ± 0.33 a |
| **γ-decalactone (µg/L)** | 88 [44] | **7.41 ± 0.60 a** | **11.08 ± 0.76 b↑** | **16.48 ± 2.41 c↑** |
| **FURANIC COMPOUNDS** | | | | |
| **Furaneol (µg/L)** | 5 [48] | **8.61 ± 1.36 a** | **21.45 ± 3.28 b↑** | **22.34 ± 4.58 b↑** |
| **Ethyl 2-furoate (µg/L)** | 16,000 [44] | **0.94 ± 0.06 b** | **0.73 ± 0.05 a↓** | **0.60 ± 0.20 a↓** |
| **Hydroxymethylfurfural (µg/L)** | 10,000 [44] | **124.30 ± 37.08 a** | **322.86 ± 11.70 b↑** | **271.86 ± 57.42 b↑** |
| **VOLATILE PHENOLS** | | | | |
| Phenol (µg/L) | - | 1.44 ± 0.29 a | 1.60 ± 0.34 a | 1.53 ± 0.38 a |
| **4-vinylphenol (µg/L)** | 35 [49] | **96.55 ± 6.85 a** | **126.74 ± 24.53 b↑** | **131.60 ± 8.13 b↑** |
| **4-ethyl-guaiacol (µg/L)** | 33 [49] | **0.10 ± 0.02 b** | **0.06 ± 0.03 a↓** | **0.07 ± 0.00 a↓** |
| **4-vinylguaiacol (µg/L)** | 40 [49] | **140.70 ± 10.88 a** | **171.47 ± 36.08 ab↑** | **184.61 ± 4.21 b↑** |

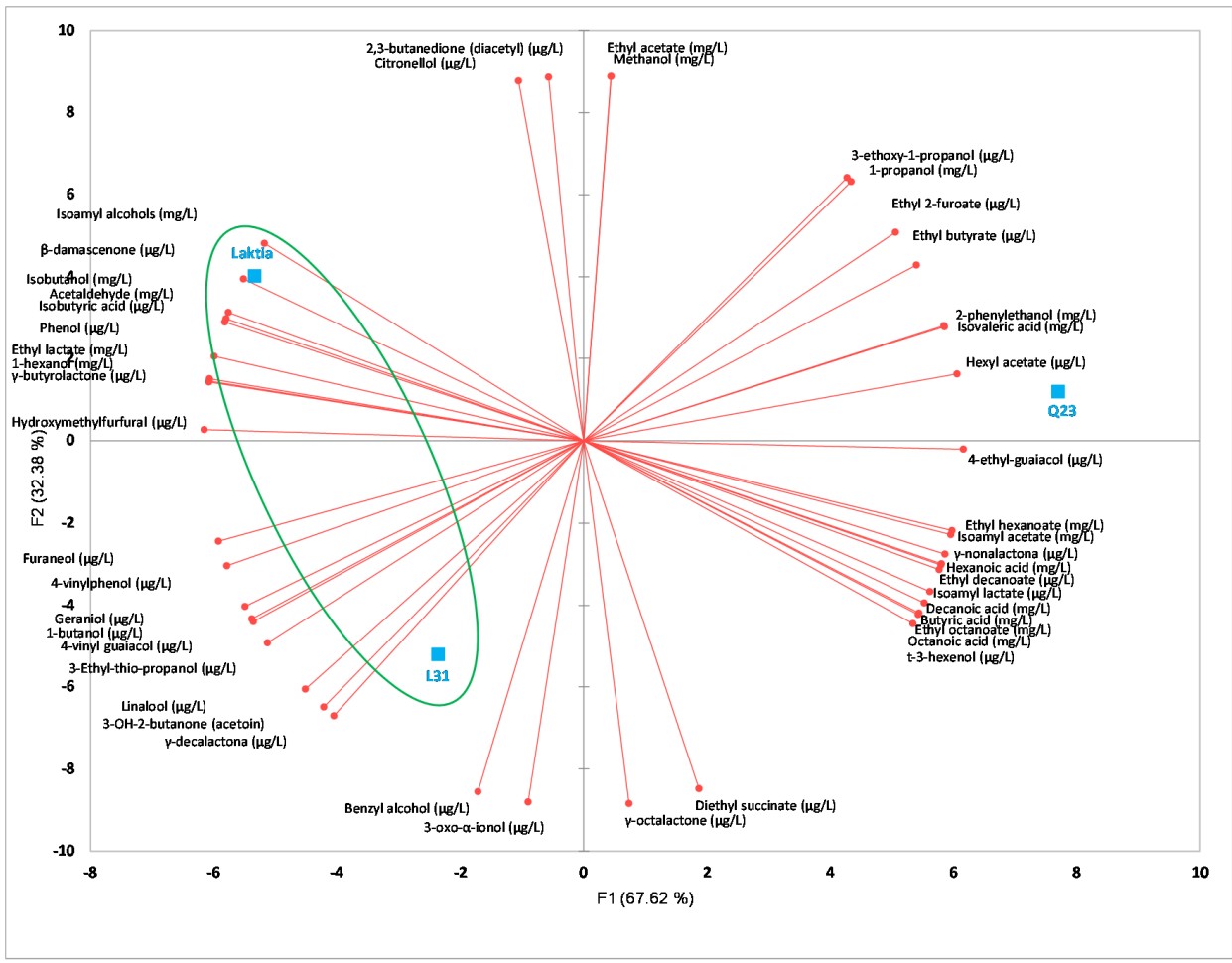

**Figure 2.** Principal component analysis (PCA) of the fermentative volatile compounds.

### 3.4. Sensory Analysis

A sensory analysis (Figure 3) was carried out to evaluate the three wines. Generally, there were no major differences, but there were significant differences in parameters such as the herbaceous aromas, where L31 obtained the highest score with 2.5 ± 0.8, while the fresh fruit parameter generated inequalities, with the lowest score for Q23 (1.3 ± 0.9) and the highest score for L31 (2.3 ± 1.0).

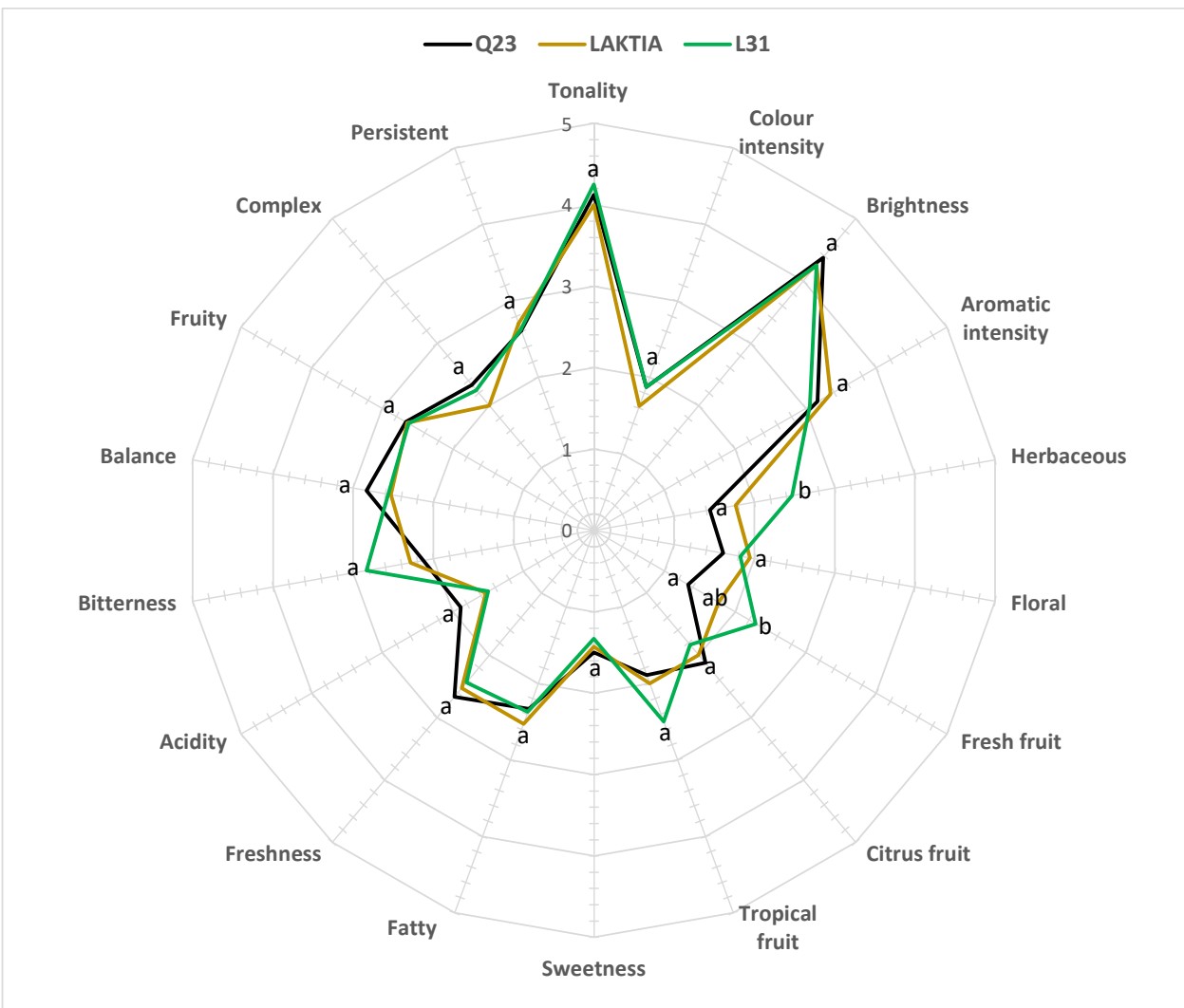

**Figure 3.** Sensory analysis of wines. The values are the average from nine tasters. The same attributes with the same letter are not significantly different $\alpha = 0.05$.

## 4. Discussion

In this research, the difference between chemical acidification using tartaric acid and biological acidification using the specific yeasts of *L. thermotolerans* was studied at a pilot-scale [11,50]. For this purpose, different oenological parameters were controlled and it was observed that the yeasts had a similar fermentation, except for the Laktia, which needed three days more to finish its fermentation [20]. It was also observed that the Laktia generated more propane-1,2,3-triol, which is characteristic of this species [23]. Both Laktia and L31 produced a lower alcoholic strength than Q23 because these yeasts use part of the sugars to create lactic acid and therefore reduce the alcohol by 0.3 to 0.7% *v/v* [12,28]. The acetic acid produced was low in all fermentations and especially in the *L. thermotolerans*, which never exceeded 0.3–0.5 g/L [15,23]. The pH was similar between Laktia and Q23, probably due to the addition of 1.5 g/L to the starting must fermented with Q23 [51], but the pH of the fermentation with L31 was slightly higher, certainly because of a lower production of lactic acid. It was also observed that despite the addition of 1.5 g/L of tartaric acid to the must fermented with the Q23 yeast, there was a large loss at the end of the tartaric stabilisation, with the three duplicates having the same amount of this acid at the end. The probability of this occurring is determined by alcohol, temperature, pH, and ionic strengths, and there may be means to predict this [52]. This loss of acidity influences the pH and total acidity due to the dissociation of organic acids; these acids influence

the pH depending on the relative strength and concentration [50]. This parameter can be improved using carboxymethylcellulose (CMC) [53] or electrodialysis with little impact on the wines [54], but biological acidification through *L. thermotolerans* has the advantage of not having to use it.

Regarding the colour parameters, it was found that the fermentation with Laktia had the lowest value of catechins, probably due to the fact that this polyphenolic compound is sensitive to small variations in acidity [55]. These flavonoid compounds, together with epicatechins, tannins, and anthocyanins [56] are easily oxidised, generating the darkening of the wine. With a lower quantity of these compounds and a similar protection, an increase in the b* parameter was observed, which correlated with a higher absorbance at 420 nm, as observed in the Laktia yeast [57]. The Folin–Ciocalteu index was also carried out [58] to determine the number of total polyphenols that the wines had. The L31 and Q23 probably had a greater amount of $SO_2$, which interfered with these measurements [59]. For more information on the visible colour spectrum, CIELab [60] and absorbance measurements at 420 nm were used; although there was a slightly significant difference, the colour of the three was practically the same.

As for volatiles, ethyl acetate was above its perception threshold of 12 mg/L [49] for all yeasts. Regarding hexyl acetate, whose olfactory descriptors are banana/apple/pear/cherry and whose perception threshold is 1.5 mg/L, although there was a big difference between *L. thermotolerans* and *S. cerevisiae*, this was below the perception threshold [41,46,61]. Although the perception threshold of ethyl lactate with acidic and fruity aromas is much higher than that produced by yeasts, a higher production was seen in *L. thermotolerans* yeasts [40,43]. Ethyl decanoate would be perceptible at Q23 and L31 as they were above the threshold of perception, which is 0.2 mg/L, and their aroma is sweet/dry fruit [41,46]. The sum of all the higher alcohols was slightly above 300 mg/L, which could have a negative effect by giving winey and irritating aromas [46,62]. Isoamyl alcohol was well above the detection threshold of 30–60 mg/L, which could produce nail polish, fuselage, and herbaceous aromas [41,42]. The perception threshold for isobutanol is 40 mg/L; Laktia was at the limit of the perception that generates fuselage aromas [46]. Moreover, 3-ethoxy-1-propanol was above the perceptible threshold (0.1 mg/L) in all cases, which can give a blackcurrant odour [42,63].

All the carbonyl compounds analysed were below the perception threshold as acetaldehyde needs 110 mg/L to be detected, with overripe apple/cut grass aromas; diacetyl has a perception threshold between 0.2 and 2.8 mg/L, with butter/caramel aromas, and acetoin has a perception threshold of 150 mg/L and a dairy/fatty odour [11,42,64]. Organic acids are important compounds as they can generate lactic, fatty, and even rancid aromas and have antioxidant, antimicrobial, and chelating functions [65,66]. All of them decreased, except isobutyric acid, which has a perception threshold of 2300 µg/L, although it increased in *L. thermotolerans* yeasts. In no case would its acid/cheese aroma be perceptible [49], but it could add more complexity to the wine [40]. There were no significant differences between norisoprenoids; they are compounds whose threshold of perception is very low [67].

In terpenes, geraniol stands out for doubling in the *L. thermotolerans* yeasts but was still far from being detected as its threshold of perception is at 20 µg/L, with flower/sweet/citrus odours [49,68]. Lactones are compounds that are produced in greater quantities when the wine interacts with the barrel, as in this case, and although γ-butyrolactone and γ-decalactone doubled, their threshold of perception was far from 20–35 mg/L for γ-butyrolactone (caramel, coconut aroma) and 88 µg/L for γ-decalactone (sweet, fruity aroma) [43,44,69]. Of the three fumaric compounds, only furaneol could be detected as its threshold is above 5 µg/L, with odours of caramel and cotton candy. The other two compounds, ethyl 2-furoate and hydroxymethylfurfural, have a threshold above 10 mg/L, with no detectable odour in these fermentations [44,48]. Finally, of the volatile phenolic compounds, 4-vinylphenol and 4-vinylguaiacol were detectable, as their detection threshold is above 35 µg/L, with clove/phenol/medicinal odours [41,49].

Finally, the sensory analysis did not show any major differences between the three wines, but the tasters detected herbaceous aromas in one of the yeasts; this aroma is characteristic of six-carbon alcohols (1-hexanol, trans-2-hexenol, trans-3-hexenol, and cis-3-hexenol) and aldehydes (1-hexanal and trans-2-hexenal) which are characterised by a green/herbaceous aroma [70], but in this case this does not correlate with the data of the volatiles analysed. As for the fresh fruit, they also detected a difference because the L31 and, to a lesser extent, the Laktia generates sensations of fresh wines; a mixture of acidity and freshness is very characteristic of this species [11,18]. Although there were no significant differences regarding the acidity parameters, it is worth mentioning that the tasters detected a slightly higher acidity in the fermentation with Q23 that had received a dose of tartaric acid prior to fermentation. This may be because acidification with tartaric acid generates sour sensations [21], as opposed to lactic acidification which tends to give more citric acid-like sensations [71].

## 5. Conclusions

The results obtained show that 1.5 g of tartaric acid is equivalent to a similar total acidity that can be obtained using *L. thermotolerans* and that this acidification would have been greater if the non-*Saccharomyces* yeasts had been allowed to grow even more. In addition, the sensory profile was found to be positive in general, with fewer short chain saturated fatty acids allowing other aromatic esters to be highlighted, and with an increase in terpenes such as geraniol. The use of a bio-tool such as *L. thermotolerans* in warm areas can be a great alternative to chemical acidification.

**Supplementary Materials:** The following are available online at https://www.mdpi.com/article/10.3390/fermentation7030193/s1, Table S1: Parameters of the Airén must after settling (the same must for the six wines).

**Author Contributions:** P.M.I.-C., A.M.-M. and L.M.-C. experimental work; C.V. and A.M.; literature review, writing, and editing; C.V. and A.M.; image design; L.M.-C. and J.M.H.; critical reading; J.M.H. and A.M. conceptualisation and experimental design. All authors have read and agreed to the published version of the manuscript.

**Funding:** Ministerio de Ciencia, Innovación y Universidades project: RTI2018-096626-B-I00 and European Regional Development Fund (ERDF), through the National Smart Growth Operational Programme FEDER INTERCONECTA EXP-00111498/ITC-20181125, project: FRESHWINES.

**Institutional Review Board Statement:** Not applicable.

**Informed Consent Statement:** Not applicable.

**Conflicts of Interest:** The authors declare that there is no conflict of interest.

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
