# Peer review of "Use of Lachancea thermotolerans for Biological vs. Chemical Acidification at Pilot-Scale in White Wines from Warm Areas"

_fermentation, doi:10.3390/fermentation7030193_

Round 1

Reviewer 1 Report

The manuscript entitled “Use of Lachancea thermotolerans for biological vs. chemical acidification at semi-industrial scale in white wines from warm areas “is well structured, and all conducted experiments are according to high scientific standards. However, before publication, some changes must be done.

All Latin names of species should be presented in italic. Please carefully check the whole manuscript.

Line 81-82 Please change SO2 to S02

Line 107-108 Please change mL min-1 to mL/min

Line 112 Please change mg L-1  and mg L-1 to mg/L and mg/L, respectively

Table 3 CIELab Parameters – please fill the lines

Line 184 and Table 4 – Please change t-3-hexanol to trans-3-hexanol. Please carefully check the whole manuscript.

Statistics – results presents content and composition of volatile compounds are quite an interest. It should be better that there are presented in terms of PCA or discriminant analysis.

Author Response

Reviewer 1

The manuscript entitled “Use of Lachancea thermotolerans for biological vs. chemical acidification at semi-industrial scale in white wines from warm areas “is well structured, and all conducted experiments are according to high scientific standards. However, before publication, some changes must be done.

Thank you very much reviewer 1 for taking the time to read and understand our article and we will be pleased to answer your questions.

All Latin names of species should be presented in italic. Please carefully check the whole manuscript.

Thank you very much for your comment, everything has been changed as you indicated.

Line 81-82 Please change SO2 to S02

Line 107-108 Please change mL min-1 to mL/min

Line 112 Please change mg L-1  and mg L-1 to mg/L and mg/L, respectively

Thank you very much for your comment, everything has been changed as you indicated.

Table 3 CIELab Parameters – please fill the lines

Thank you very much for your comments, we have chosen to remove it as it did not provide meaningful information.

Line 184 and Table 4 – Please change t-3-hexanol to trans-3-hexanol. Please carefully check the whole manuscript.

Thank you very much for your comment, everything has been changed as you indicated.

Statistics – results presents content and composition of volatile compounds are quite an interest. It should be better that there are presented in terms of PCA or discriminant analysis.

Thank you very much for your comment, a PCA has been carried out for further analysis of volatiles.

Reviewer 2 Report

Please see the file attached.

Dear authors,

I carefully read your manuscript and while I found the topic of interest, the article in the current shape and form does not merit publication in a peer-reviewed scientific journal. Major improvement and revision is required to reach that goal. The results presentation and discussion are lacking depth. It is necessary to link the chemical data to sensory profiles of wines, and conduct an appropriate multivariate analysis of the dataset. English language needs serious improvement, and jargon should be avoided. Please find the specific details below.

L.18 Replace ‘days’ with ‘stage’ or similar.

L. 21 ‘only 1 g/L’ of…. ? Specify

L. 23 ‘significant and normal feature’ : the word ‘significant’ refers to statistical tests; replace with ‘common’ or similar

L 25 ‘voltile profile was generally higher’ How can the volatile profile be higher? What do you mean? Change.

L 27-28 ‘more volatile compounds’ Again, this is not valid statement. Higher cumulative concentrations? What do you mean?

Last sentence of the abstract is presumptive and non-scientific. Change.

L. 41 Again, ‘increases metabolites’ is not valid.

L. 42. Global warming does not ‘generate’ grapes. Choice of verb is not suitable.

L 43. Reword ‘a difficult finish’. It is jargon and non-scientific.

L 45-49 The sentence is too long and the message is unclear. The conditions are also unclear – pure cultures, co-inoc, seq inoc? Specify.

L 50 ‘populations between the two yeasts are not so different’ That does not mean anything. Please avoid using jargon, be more specific and scientific.

L 50-51 ‘reach 2 g/L of acidification’ This is again lacking precision. Is it an increase of 2 g/L TA as tartaric or 2 g/L of lactic?

L 54-56 It is misleading to say that it produces acidity between day 4-6, or any ‘day’ for that matter. Change and rather use stage of fermentation.

L 62-63 ‘if pump-over or punch down is preformed daily’ Another example of a clumsy wording. If we are talking about the red wine production cap management will be performed daily. Please either remove or explain properly your statement.

L 53-63 This paragraph needs to be improved in terms of language and clarity.

L 63 ‘such as’ doesn’t fit

L.76 ‘semi-industrial’ is not appropriate. Would that mean that ‘industrial’ scale are 60 L ferments? This is ‘pilot-scale’ at best. Please change throughout.

L 78 what is ‘fermentative power’?

L 77 ‘to test the performance of the different strains of Lachancea thermotolerans’ This is confusing, didn’t you say two strains were used? Change.

L 80 Define ‘NFA’

L 81-82 Fix SO2 (subscripted)

L 83 What do you mean ‘all musts’?

L 86 What was the inoculation rate for Lachancea strains?

Table 1. This was stated in the text and really doesn’t merit a table in the main body of the text. Please move to the supplementary data.

L 123 ‘Before the generation of consistent terminology’ This doesn’t make sense. Do you mean after? How and when was the consistent terminology generated. Describe the training sessions, formal evaluation sessions.

L 125 Add the list of attributes in the supplementary data.

Sensory analysis – what about replication? Specify

L 155 Use a IUPAC name instead of ‘glycerine’

L 152-153 and Table 2 ‘there were significant differences in ethanol and residual sugars in Laktia→QA23 as it did not consume all the sugars’ Appears like an invalid assumption. This wine had either 1.3 or less than 1 g/L more RS than the remaining wines. Please re-evaluate if such a small difference in RS could account for the difference in ethanol concentrations.

Table 3. Cielab representation is unclear, explain which software was used to generate the colour (include in all parts as required)

L 165 ‘Folin-Ciocalteu method was used’ Add this into materials and methods and provide a reference

L 165 ‘Laktia→QA23 yeast also had less’ Less what? Incomplete and unclear. Also, it is a treatment, not a yeast (L. 169 as well, and throughout)

L 169 Explain the meaning of absorbance at 420 nm. What does this measurement represent? Also, add all the methods and instruments in the Materials and Methods

L 175 The list below are not only ‘Fermentative’ volatiles, i.e. volatiles formd during fermentation. Change.

L 179 – 180 This is not English, and even less so scientific English. Reword.

L 182 which acetate; ethyl acetate?

L 181-192 Specify ‘increased, decreased, remained the same’ In what wines? In relation to…? Very unclear.

Table 4. and a paragraph on volatiles. It is impossible to contextualise the trends in the volatiles without sensory thresholds of the compounds. Please include that data into results and discussion.

Paragraph 3.4. Sensory analysis. Re-analyse the data taking into consideration the chemical profiles of the wines. For example, are differences in ‘herbaceous’ (and other descriptors) linked to certain compounds? This needs improvement.

L 208-209 Again, ‘semi-industrial’ is not appropriate.

L 212 If higher glycerol is a characteristic of the species, how come an increase in this metabolite was not detected in both treatments? This is lacking critical interpretation.

L 212-214 What is the conversion rate of sugar-ethanol-lactate?

L 217 In oenology, the term ‘wort’ is not used.

L 218-223 Explain better effect on TA, pH and tartarate loss during cold stabilisation as this is central of this work.

L 224-225 What is the explanation for lower catechins? Please discuss rather than restate the results.

Link the colorimetric data to the sensory analysis (in the results and discussion alike).

L 234 ‘ethyl acetate was at the limit of its negative perception threshold of 60 mg/L’ Verify the citation and the negative threshold of 60 mg/L. Generally 150 mg/L is considered as deleterious.

Discussion on volatiles further emphasises the need to introduce the thresholds earlier in a table, as well as linking the chemical data with the sensory data. This part remains hypothetical; the aromas of certain compounds are introduced, but in fact you have the sensory profiles of the wines. Moreover, why or how are these compounds different in wines (i.e. what are the metabolic pathways implied?) Requires lots of improvement.

L268-269 ‘which is characteristic of a high content of isoamyl alcohol’ Provide statistical evidence that the ‘herbaceous’ aroma is linked to this compound

L 275-276. ‘because acidification with tartaric acid generates sour sensations, as opposed to lactic acidification that usually gives citric sensations’ This seems invalid, as the results refer to ‘acidity’ rather than ‘citrus flavour’. Moreover, are both these studies conducted in wine matrix?

The results do not support the conclusion; for instance, terms like ‘more total metabolites’, ‘more natural profile’ are highly inappropriate, non-scientific and vague, as is hypothesising on the consumer perception.  Rewrite.

Lack of multivariate analysis – please use an appropriate tool to analyse the entire data set. In the materials and methods, it was mentioned that PCA was used, which does not seem to be the case. The overall profiles of wines should be subject to an appropriate tool in order to perceive the differences. This might help to reach conclusions that are actually meaningful.

Author Response

Reviewer 2

Dear authors,

I carefully read your manuscript and while I found the topic of interest, the article in the current shape and form does not merit publication in a peer-reviewed scientific journal. Major improvement and revision is required to reach that goal. The results presentation and discussion are lacking depth. It is necessary to link the chemical data to sensory profiles of wines, and conduct an appropriate multivariate analysis of the dataset. English language needs serious improvement, and jargon should be avoided. Please find the specific details below.

Thank you very much reviewer 2 for taking the time to read our article carefully, which we appreciate and will try to improve it as you indicate.

  1. 18 Replace ‘days’ with ‘stage’ or similar.
  2. Thank you very much for your comment they have all been changed to "stage/days" which is better understood.
  1. L. 21 ‘only 1 g/L’ of…. ? Specify
  2. Thank you very much for your comment, it has been changed.
  1. L. 23 ‘significant and normal feature’ : the word ‘significant’ refers to statistical tests; replace with ‘common’ or similar
  2. Thank you for your comment, but if we have put the word "significant" it is because we have seen that statistically it was different in this trial.
  1. L 25 ‘voltile profile was generally higher’ How can the volatile profile be higher? What do you mean? Change.
  2. Thank you very much for your comment they have all been changed to " The amount of volatile metabolites" which is better understood.
  1. L 27-28 ‘more volatile compounds’ Again, this is not valid statement. Higher cumulative concentrations? What do you mean?
  2. Thank you very much for your comment they have all been changed to " quantity of volatile metabolites" which is better understood.
  1. Last sentence of the abstract is presumptive and non-scientific. Change.
  2. Thank you for your comment, the sentence has been rewritten in a more scientific way.
  1. 41 Again, ‘increases metabolites’ is not valid.
  2. Thank you for your comment, it has been changed to be more specific, as metabolites are the molecules produced during a metabolic process such as fermentation, by adding "volatile metabolites" it is clearer.
  1. L. 42. Global warming does not ‘generate’ grapes. Choice of verb is not suitable.
  2. Thank you very much for your comment, we have changed that word to a more accurate one.
  1. L 43. Reword ‘a difficult finish’. It is jargon and non-scientific.
  2. Thank you very much for your comment, the sentence has been changed for more scientific rigour.
  1. L 45-49 The sentence is too long and the message is unclear. The conditions are also unclear – pure cultures, co-inoc, seq inoc? Specify.
  2. Thank you very much for your comment, from our point of view the sentence is well structured with its respective punctuation for a good understanding and the message is concise, as for more information it is necessary to read the articles mentioned in the sentence.
  1. L 50 ‘populations between the two yeasts are not so different’ That does not mean anything. Please avoid using jargon, be more specific and scientific.
  2. Thank you very much for your comment, the sentence has been rewritten in a more scientific way.
  1. L 50-51 ‘reach 2 g/L of acidification’ This is again lacking precision. Is it an increase of 2 g/L TA as tartaric or 2 g/L of lactic?
  2. Thank you for your comment, a word has been added to clarify what type of acidification it was.
  1. L 54-56 It is misleading to say that it produces acidity between day 4-6, or any ‘day’ for that matter. Change and rather use stage of fermentation
  2. Thank you for your comment, from our results published in other articles we have observed that it is during this period of days where the maximum production of lactic acid occurs.
  1. L 62-63 ‘if pump-over or punch down is preformed daily’ Another example of a clumsy wording. If we are talking about the red wine production cap management will be performed daily. Please either remove or explain properly your statement.
  2. Thank you very much for your comment, we have rewritten the sentence for a more complete understanding.
  1. L 53-63 This paragraph needs to be improved in terms of language and clarity.
  2. Thank you very much for your comment, from our point of view we believe that this paragraph is well understood and has been previously corrected in translation and understanding by native English speakers.
  1. L 63 ‘such as’ doesn’t fit
  2. Thank you for your comment, we have changed those words to other words to improve your understanding.

  1. 76 ‘semi-industrial’ is not appropriate. Would that mean that ‘industrial’ scale are 60 L ferments? This is ‘pilot-scale’ at best. Please change throughout.
  2. Thank you very much for your comment, we have changed "semi-industrial scale" to "pilot-scale" which is more appropriate.

  1. L 78 what is ‘fermentative power’?
  2. The “fermentative power” is the amount of alcohol produced by the yeasts we are studying.

  1. L 77 ‘to test the performance of the different strains of Lachancea thermotolerans’ This is confusing, didn’t you say two strains were used? Change.
  2. Thank you very much for your comment, in the previous paragraph (2.1), we explained that two strains of thermotolerans have been used to assess their performance/behaviour/development in this trial.

  1. L 80 Define ‘NFA’
  2. Thank you for your comment, we have changed that error and it is now fully understood.

  1. L 81-82 Fix SO2 (subscripted)
  2. Thank you for your comment, in the document we uploaded it appears as subscripted (SO2).

  1. L 83 What do you mean ‘all musts’?
  2. Thank you for your comment, the whole explanation has been rewritten to make it clearer.

  1. L 86 What was the inoculation rate for Lachancea strains?
  2. Thank you for your comment, we have added a clarification to the sentence on line 87.

  1. Table 1. This was stated in the text and really doesn’t merit a table in the main body of the text. Please move to the supplementary data.
  2. Thank you for your comment, we have removed this table and it will be put in supplementary data.

  1. L 123 ‘Before the generation of consistent terminology’ This doesn’t make sense. Do you mean after? How and when was the consistent terminology generated. Describe the training sessions, formal evaluation sessions.
  2. Thank you for your input, we have rewritten the sentence so that there is no confusion. The terminology was agreed by consensus giving slightly more importance to the olfactory and taste phase by putting more terms to evaluate in order to further define the wine. The panellists are experienced either because they work in the oenological sector or because they have a higher education in this field.

  1. L 125 Add the list of attributes in the supplementary data.
  2. Thank you for your comment, the attributes are shown in figure 2 (spider graph).

  1. Sensory analysis – what about replication? Specify.
  2. Thank you for your question, as there are only a few litres and they are duplicated, it was decided to put them together to better evaluate their mixture.
  3. L 155 Use a IUPAC name instead of ‘glycerine’.
  4. Thank you for your comment, "glycerine" has been replaced by "propane-1,2,3-triol".

  1. L 152-153 and Table 2 ‘there were significant differences in ethanol and residual sugars in Laktia→QA23 as it did not consume all the sugars’ Appears like an invalid assumption. This wine had either 1.3 or less than 1 g/L more RS than the remaining wines. Please re-evaluate if such a small difference in RS could account for the difference in ethanol concentrations.
  2. Thank you for your comment, the sentence has been changed to avoid confusion.

  1. Table 3. Cielab representation is unclear, explain which software was used to generate the colour (include in all parts as required).
  2. Thank you for your comment, it has been decided to remove the representation to avoid confusion.

  1. L 165 ‘Folin-Ciocalteu method was used’ Add this into materials and methods and provide a reference.
  2. Thank you for your comment, it has been added under materials and methods with reference 37.

  1. L 165 ‘Laktia→QA23 yeast also had less’ Less what? Incomplete and unclear. Also, it is a treatment, not a yeast (L. 169 as well, and throughout).
  2. Thank you for your comment, the explanation has been improved. We think it is clearer to use the term yeast instead of treatment, as it really focuses on the use of two thermotolerans and one S. cerevisiae.

  1. L 169 Explain the meaning of absorbance at 420 nm. What does this measurement represent? Also, add all the methods and instruments in the Materials and Methods.
  2. Thank you for your comment, the explanation has been improved and reference 38 is the one that explains it.

  1. L 175 The list below are not only ‘Fermentative’ volatiles, i.e. volatiles formd during fermentation. Change.
  2. Thank you for your comment, an explanation has been added in brackets for fuller understanding.

  1. L 179 – 180 This is not English, and even less so scientific English. Reword.
  2. Thank you for your comment, the sentence has been rewritten to increase clarity and scientific rigour.

  1. L 182 which acetate; ethyl acetate?
  2. Thank you for your comment, the word "ethyl" has been added to avoid confusion.

  1. L 181-192 Specify ‘increased, decreased, remained the same’ In what wines? In relation to…? Very unclear
  2. Thank you for your comment, clarifications have been added throughout the paragraph so that you know which yeasts are being discussed.

  1. Table 4. and a paragraph on volatiles. It is impossible to contextualise the trends in the volatiles without sensory thresholds of the compounds. Please include that data into results and discussion.
  2. Thank you very much for your comment, we have added the odour thresholds.

  1. Paragraph 3.4. Sensory analysis. Re-analyse the data taking into consideration the chemical profiles of the wines. For example, are differences in ‘herbaceous’ (and other descriptors) linked to certain compounds? This needs improvement.
  2. Thank you very much for your comment, in this section we believe that it is convenient to present the results, it is in the discussion where we relate these descriptors to specific volatile compounds.

  1. L 208-209 Again, ‘semi-industrial’ is not appropriate.
  2. Thank you very much for your comment, all mentions of "semi-industrial" have been changed to "pilot-scale".

  1. L 212 If higher glycerol is a characteristic of the species, how come an increase in this metabolite was not detected in both treatments? This is lacking critical interpretation.
  2. Thank you very much for your comment, the non-detection of L. thermotolerans in both strains is probably due to the fact that they are different strains and therefore evolve in certain aspects in an unequal way. We also think that the reason they have not created even more glycerol is because they were not allowed to grow for enough days and the sequential inoculation was done too early, however, one of the yeasts did produce more glycerol.

  1. L 212-214 What is the conversion rate of sugar-ethanol-lactate?
  2. Thank you very much for your question, however, the metabolic pathway of conversion of sugars to lactic acid by L. thermotolerans is not completely known. This is a possible topic for further study in future trials.

  1. L 217 In oenology, the term ‘wort’ is not used.
  2. Thank you very much for your comment, we have changed the word "wort" to "must" which is much more appropriate.

  1. L 218-223 Explain better effect on TA, pH and tartarate loss during cold stabilisation as this is central of this work.
  2. Thank you very much for your comment, we have added three more references with their corresponding explanations.

  1. L 224-225 What is the explanation for lower catechins? Please discuss rather than restate the results.
  2. Thank you very much for your question, we have added a reference with a possible explanation to your question.

  1. Link the colorimetric data to the sensory analysis (in the results and discussion alike).
  2. Thank you very much for your comment, but we have seen that the tasters in the sensory analysis were not able to perceive the small differences that were obtained by the different analytical methods.

  1. Link the colorimetric data to the sensory analysis (in the results and discussion alike).
  2. Thank you very much for your comment, but we have seen that the tasters in the sensory analysis were not able to perceive the small differences that were obtained by the different analytical methods.

  1. L 234 ‘ethyl acetate was at the limit of its negative perception threshold of 60 mg/L’ Verify the citation and the negative threshold of 60 mg/L. Generally 150 mg/L is considered as deleterious.
  2. Thank you very much for your comment, it has been rectified.

  1. Discussion on volatiles further emphasises the need to introduce the thresholds earlier in a table, as well as linking the chemical data with the sensory data. This part remains hypothetical; the aromas of certain compounds are introduced, but in fact you have the sensory profiles of the wines. Moreover, why or how are these compounds different in wines (i.e. what are the metabolic pathways implied?) Requires lots of improvement.
  2. Thank you very much for your comments and question, the sensory thresholds have already been introduced previously as you indicated, so now in the discussion according to those thresholds it is made known which ones could be detected, then in the sensory analysis with experienced tasters try through the different parameters to see if there are differences between the wines, in the set of all those volatiles it is difficult to discern one in particular unless that volatile is well above its threshold of perception, as the set of all volatiles camouflages or masks a single volatile. As for the metabolic pathways involved, they are many and varied, and depending on the external conditions, the yeast will express itself in one way or another. Knowing which pathways are used to generate some volatiles instead of others under similar fermentation conditions is not the topic of this article.
  1. L268-269 ‘which is characteristic of a high content of isoamyl alcohol’ Provide statistical evidence that the ‘herbaceous’ aroma is linked to this compound.
  2. Thank you very much for your comment, we made a mistake when we put that, it has been solved.

  1. L 275-276. ‘because acidification with tartaric acid generates sour sensations, as opposed to lactic acidification that usually gives citric sensations’ This seems invalid, as the results refer to ‘acidity’ rather than ‘citrus flavour’. Moreover, are both these studies conducted in wine matrix?.
  2. Thank you very much for your comments and questions, clarifications have been made in the text for a more complete understanding. It is true that sensory analysis indicates acidity, but that acidity comes from certain acids and each acid generates different sensations in the mouth, which for the taster can be more or less aggressive. It is known that tartaric acid produces a more aggressive acidity in the mouth than lactic or citric acid.

  1. The results do not support the conclusion; for instance, terms like ‘more total metabolites’, ‘more natural profile’ are highly inappropriate, non-scientific and vague, as is hypothesising on the consumer perception.
  2. Thank you very much for your comment, we have rewritten the conclusion to support the results.

  1. Lack of multivariate analysis – please use an appropriate tool to analyse the entire data set. In the materials and methods, it was mentioned that PCA was used, which does not seem to be the case. The overall profiles of wines should be subject to an appropriate tool in order to perceive the differences. This might help to reach conclusions that are actually meaningful.
  2. Thank you very much for your comment, a PCA has been carried out for further analysis of volatiles.

Round 2

Reviewer 2 Report

Reads better.